# Mechanical and Acoustic Properties of Alloys Used for Musical Instruments

**DOI:** 10.3390/ma15155192

**Published:** 2022-07-26

**Authors:** Mariana Domnica Stanciu, Mihaela Cosnita, Constantin Nicolae Cretu, Horatiu Draghicescu Teodorescu, Mihai Trandafir

**Affiliations:** 1Faculty of Mechanical Engineering, Transilvania University of Brasov, B-dul Eroilor 29, 500036 Brasov, Romania; draghicescu.teodorescu@unitbv.ro (H.D.T.); mihai.trandafir96@gmail.com (M.T.); 2Department of Product Design Mechatronics and Environment, Transilvania University of Brasov, B-dul Eroilor 29, 500036 Brasov, Romania; 3Department of Electrical Engineering and Applied Physics, Transilvania University of Brasov, B-dul Eroilor 29, 500036 Brasov, Romania; cretu.c@unitbv.ro

**Keywords:** aluminum, stainless steel, elastic properties, tensile test, intrinsic transfer matrix, triangle

## Abstract

Music should be integrated into our daily activities due to its great effects on human holistic health, through its characteristics of melody, rhythm and harmony. Music orchestras use different instruments, with strings, bow, percussion, wind, keyboards, etc. Musical triangles, although not so well known by the general public, are appreciated for their crystalline and percussive sound. Even if it is a seemingly simple instrument being made of a bent metal bar, the problem of the dynamics of the musical triangle is complex. The novelty of the paper consists in the ways of investigating the elastic and dynamic properties of the two types of materials used for musical triangles. Thus, to determine the mechanical properties, samples of material from the two types of triangles were obtained and tested by the tensile test. The validation of the results was carried out by means of another method, based on the modal analysis of a ternary system; by applying the intrinsic transfer matrix, the difference between the obtained values was less than 5%. As the two materials behaved differently at rupture, one having a ductile character and the other brittle, the morphology of the fracture surface and the elementary chemical composition were investigated by scanning electron microscopy (SEM) and analysis by X-ray spectroscopy with dispersion energy (EDX). The results were further transferred to the finite element modal analysis in order to obtain the frequency spectrum and vibration modes of the musical triangles. The modal analysis indicated that the first eigenfrequency differs by about 5.17% from one material to another. The first mode of vibration takes place in the plane of the triangle (transverse mode), at a frequency of 156 Hz and the second mode at 162 Hz, which occurs due to vibrations of the free sides of the triangle outside the plane, called the torsion mode. The highest dominant frequency of 1876 Hz and the sound speed of 5089 m/s were recorded for the aluminum sample with the ductile fracture in comparison with the dominant frequency of 1637 Hz and the sound speed of 4889 m/s in the case of the stainless steel sample, characterized by brittle fracture.

## 1. Introduction

Music listening should be integrated into our daily life activities due to its great effects on human holistic health. The sounds emitted by percussion instruments, such as triangles, are used in various therapies because it has been found that the emitted frequencies stimulate cerebral circulation.

There are reports [1,2] on the music potential in promoting the neuroplasticity through dopamine stimulation, thus improving the cognitive factor, emotions and motivation [3]. Alloys are used in the construction of percussion and wind instruments, and have been used since ancient times. There are numerous studies on the constructive evolution of different instruments that contain metallic elements whose impact produces highly appreciated musical sounds [4,5,6,7,8]. Currently, many constructive forms of percussion instruments have been developed, based on systematic studies on the theory of musical sounds and the physical phenomena of sound production with implications for music. Among these modern tools, we can mention: the profiled chime; the pentangle; new metallic musical instruments (the aluphone, the sixxen and the percussion instruments invented by Melchiorre). The musical role of the triangle in the orchestra began with the Turkish music from the 18th century, the instrument being later used by Franz Liszt in the third part of the Piano Concerto no. 1 in E flat major, known as the Triangle Concerto [4].

The triangle is obtained by bending a metal bar with a circular section, in the form of an open triangle, with approximately equal sides [9,10,11]. Although the geometric shape seems relatively simple, the triangle is a percussion instrument with a relatively complex structure, whose acoustics depend on the material from which it is made. The sound is produced by hitting the sides with a metal rod, the triangle being supported by an elastic thread [12,13,14]. The musical interpretation techniques aim at the intensity of the beats, the timing and the damping of the vibrations by the percussionist, in accordance with the rhythm and the musical measures in the concert [5,15,16,17]. Longitudinal and transverse vibrations produced by hitting the triangle propagate along the entire length of the kinked bar, generating different eigenmodes. Thus, for the longitudinal vibrations, it can be assumed that the frequency of a bar of circular section and length *L* is calculated from the relation (1), according to [11]:(1)$$f_{n}=\frac{a}{2}{\left(2k+1\right)}^{2}\frac{\pi }{8L^{2}}\sqrt{\frac{E}{\rho }},$$
where $f_{n}$ is the natural frequency (Hz), a—the radius of the circular section (mm); *k*—the number of vibration modes, *L*—the length of the bar (mm); *E*—longitudinal elasticity modulus (N/mm^2^), ρ–density (kg/m^3^).

Research on the vibrations of the triangle has shown that transverse modes have small displacements and similar to the vibration modes of a straight bar [11].

Thus, one of the hypotheses states that there are no differences between the vibrations of a straight bar and those of the bent bar. However, numerical studies have shown that the distribution of stresses in the curved bar differs from that of the right bar, in the area where it is bent, which changes the ways of propagation of vibrations in the material. Thus, in the bent areas the characteristic impedance differs from that of the right bar due to the geometric discontinuity [5,15,16,17]. According to relation (1), an important parameter in the formation of eigenfrequencies are the mechanical proprieties of material, through the longitudinal modulus of elasticity. Thus, the most used materials for the triangle are steel, aluminum, high quality bronze alloys and copper–beryllium alloys.

The studies carried out on the materials used in the construction of idiophone instruments, such as the musical triangles, are relatively scarce, the paper thus highlighting the mechanical properties of steel and aluminum used for triangles, in correlation with their eigenfrequencies. Numerous papers on the chemical composition and surface morphology of metal samples highlight different methods, such as X-ray photoelectron spectroscopy, morphology test, potentiodynamic polarization (PDP) curve test and electrochemical impedance spectroscopy (EIS) test [18,19].

In our previous study [4], the dynamic analysis of the triangles was investigated through the hammer hits on triangle sides and their signal were recorded.

The novelty of this study consists of the complementary approach in the methods applied for mechanical, acoustic and chemical characterization of the alloys used in the construction of musical instruments. Two experimental methods were herein used, namely the mechanical tensile test and the hybrid method, based on the intrinsic transfer matrix, and the results where further applied within the modal analysis in order to obtain the vibration mode of the two triangle types.

## 2. Materials and Methods

### 2.1. Materials

From the point of view of the materials from which they were produced, two types of triangles were studied, namely triangles made of stainless steel coded T1 and triangles made of aluminum coded T2 (Figure 1). As the analyzed triangles were purchased on the market for musical instruments, there was no information on the type and characteristics of the materials in the structure of the triangles. In previous studies, the dynamic characteristics of triangles were determined by using the experimental modal analysis, as indicated in [4], and afterwards the aluminum and stainless samples were extracted from the musical triangles in order to determine their mechanical properties. The mechanical properties were investigated by two different methods, tensile testing and intrinsic transfer matrix. Cylindrical samples were prepared in accordance with DIN EN ISO 6892-1, as can be seen in Figure 1. From each triangle, two material samples coded T1.1 and T1.2 for stainless steel, respectively T2.1, were extracted. and T2.2 for aluminum, as seen in Figure 1. Their physical characteristics are indicated in Table 1.

### 2.2. Characterization Techniques

#### 2.2.1. Tensile Test

The Lloyd Instruments LS100 Plus universal mechanical testing machine with a maximum load capacity of 100 kN, from the Department of Mechanical Engineering at Transilvania University of Brașov, was used for the tensile test (Figure 2). The elongation of the sample during the stress was measured with the EPSILON extensometer. The loading rate used during the test was 5 mm/min. The data acquisition was performed with the Nexygen Plus software, and the processing of the characteristic curves was accomplished in the Excel software.

#### 2.2.2. Intrinsic Transfer Matrix

The second method used to determine the longitudinal elasticity modulus of the tested samples was a non-destructive one, which consists in the modal analysis of an elastic system and which correlates the real eigenvalues with the frequencies of the eigenmode of the system. This method is known as the transfer matrix method, used to determine the wave propagation of continuous waves or finite pulses in homogeneous, inhomogeneous or multilayer elastic media [20,21,22,23,24,25]. The method is based on the construction of a system of collinear bars in which the acoustic and elastic parameters are known, the investigated sample being placed between the two calibrated bars. Thus, a ternary system is formed from three elastic rods with lengths denoted $l_{1}$, $\;l_{2}$ and $\;l_{3}$, having the acoustic impedance $Z_{1}$, $Z_{2}$ and $Z_{3}$. The rods 1 and 3 are made of identical materials. Because the standing wave is always confined only inside the system, in the case of an elastic medium of length l, the amplitudes of the Fourier components at the input and output are connected by means of the intrinsic transfer matrix, which in this simple case, involves only a simple forward and backward propagation of the wave inside the medium (Figure 3).

According to [20,21,22,23,24], the intrinsic transfer matrix denoted $T\left(\omega \right)$ of the ternary system for longitudinal wave propagation is (2):(2)$$T\left(\omega \right)=\frac{1}{4}⬝\left(\begin{array}{cc} e^{ik_{3}l_{3}} & 0\\ 0 & e^{-ik_{3}l_{3}} \end{array}\right)⬝\left(\begin{array}{cc} 1+\frac{Z_{2}}{Z_{3}} & 1-\frac{Z_{2}}{Z_{3}}\\ 1-\frac{Z_{2}}{Z_{3}} & 1+\frac{Z_{2}}{Z_{3}} \end{array}\right)⬝\left(\begin{array}{cc} e^{ik_{2}l_{2}} & 0\\ 0 & e^{-ik_{2}l_{2}} \end{array}\right)⬝\left(\begin{array}{cc} 1+\frac{Z_{1}}{Z_{2}} & 1-\frac{Z_{1}}{Z_{2}}\\ 1-\frac{Z_{1}}{Z_{2}} & 1+\frac{Z_{1}}{Z_{2}} \end{array}\right)⬝\left(\begin{array}{cc} e^{ik_{1}l_{1}} & 0\\ 0 & e^{-ik_{1}l_{1}} \end{array}\right),$$
where k is the wave number.

The eigenvalues of the matrix (2) are determined from relation (3):(3)$$\lambda_{1,2=}{\left\{{\left(\frac{Z_{1}+Z_{2}}{2\sqrt{Z_{1}Z_{2}}}\right)}^{2}⬝\cos\left(k_{1}l_{1}+k_{2}l_{2}+k_{1}l_{3}\right)-{\left(\frac{Z_{1}-Z_{2}}{2\sqrt{Z_{1}Z_{2}}}\right)}^{2}⬝\cos\left(k_{1}l_{1}-k_{2}l_{2}+k_{1}l_{3}\right)\right\}}^{2}\pm \;\sqrt{{\left\{{\left(\frac{Z_{1}+Z_{2}}{2\sqrt{Z_{1}Z_{2}}}\right)}^{2}⬝\cos\left(k_{1}l_{1}+k_{2}l_{2}+k_{1}l_{3}\right)-{\left(\frac{Z_{1}-Z_{2}}{2\sqrt{Z_{1}Z_{2}}}\right)}^{2}⬝\cos\left(k_{1}l_{1}-k_{2}l_{2}+k_{1}l_{3}\right)\right\}}^{2}}-1,$$

Consequently, the condition for real equal eigenvalue becomes (4):(4)$${\left\{{\left(\frac{\rho_{1}c_{1}+\rho_{2}c_{2}}{2\sqrt{\rho_{1}\rho_{2}c_{1}c_{2}}}\right)}^{2}⬝\cos\left(k_{1}l_{1}+k_{2}l_{2}+k_{1}l_{3}\right)-{\left(\frac{\rho_{1}c_{1}-\rho_{2}c_{2}}{2\sqrt{\rho_{1}\rho_{2}c_{1}c_{2}}}\right)}^{2}⬝\cos\left(k_{1}l_{1}+k_{2}l_{2}+k_{1}l_{3}\right)\right\}}^{2}-1=0,$$

Figure 4a graphically shows the components of the test system. Sample (1) is caught in holder (3) and excited in the longitudinal direction by the impact hammer (2). The test is based on the Doppler effects the frequencies being measured with an interferometer which allows a precise determination of the vibrational movement of the object [23,24,25,26,27]. Thus, a light beam was produced by the vibrometer 4 (OMETRON VQ-400-A) from the Laboratory of Acoustic Physics at Transilvania University of Brasov, which recorded the vibrational motion of the analyzed sample and transmitted these data via an data acquisition device (5) to a computer. For the stainless steel samples, the ternary system consisted of calibrated aluminum bars and the stainless steel sample in the middle. For the aluminum samples, the ternary system consisted of two brass bars and the aluminum sample between the two brass bars (Figure 4b).

The recorded signal was processed by applying the Fourier transform method to the Labview analysis software from which the natural frequency spectrum of the analyzed standard bar was obtained. Based on the values of the natural frequencies, the velocity of sound propagation in the materials of the ternary system were then determined, and subsequently, the values of the modulus of elasticity were calculated. The results of the two methods applied in this study were compared.

#### 2.2.3. Scanning Electron Microscopy (SEM) and Energy-Dispersive X-ray Spectroscopy (EDX) Analysis

Micrographs of samples surface morphology were obtained by using a scanning electron microscope (SEM, JP), Hitachi—Science & Technology, Berkshire, UK, S3400N, type II, and the quantitative elemental analysis of the samples was performed with EDX Thermo Electron Scientific Instruments LLC, Madison, WI, USA (Thermo, Ultra Dry, Noran System 7, NSS Model, 2,000,000 counts/s), with the sensitivity up to a few atomic percentages, from the Research—Development Institute at Transilvania University of Brasov. The samples analyzed at SEM were extracted from the breaking area of the tensile tested samples, respectively from the area of compression of the samples, as a result of the catching in the grip of the universal testing machine.

#### 2.2.4. Simulation of Modal Analysis of Triangles

The Simcenter 12 software from the Mechanical Engineering Department at Transilvania University of Brasov was used for both modal analysis and geometric model. The meshing of the model is based on three-dimensional finite elements with 4 nodes of tetrahedral shape, coded CTETRA, being obtained 1,988,972 finite elements and 38,066 nodes. In the finite element analysis, the determination of the boundary conditions was carried out by blocking the translational movements on the three axes of a node in the real clamping area of the triangle, as can be seen in Figure 5. In our previous work [4] the method and the results of the modal analysis are presented in more detail. The experimental values of the elastic characteristics of the materials were considered in this simulation.

## 3. Results and Discussion

### 3.1. Tensile Test (TT)

Figure 6 summarizes the tensile stress–strain curves of the two types of materials. Unlike aluminum, the stress–strain curve of stainless steel is nonlinear, which is in accordance with Tylek et al. and Luecke et al. [26,27], as the obtained curves are characteristic to carbon steel and stainless. Additionally, the stress–strain curve of stainless steel does not show a plateau before deformation hardening, which is observed in the stress–strain curve of aluminum (Figure 6a,c). Prominent necking behavior was present in case of aluminum samples compared to stainless steel, which indicates a brittle fracture mode (Figure 6b,d).

Following the tensile test of the materials in the structure of the triangles, it was found that the aluminum alloy has a longitudinal modulus of elasticity of 66,594 MPa, which is approximately 3 times lower than that of stainless steel. Compared to the literature, the values obtained experimentally on samples obtained from triangles show an average deviation of about 3.61% for aluminum and 1.18% for stainless steel [26,27,28,29,30]. The mechanical properties determined by the tensile test are summarized in Table 2. The aluminum sample is type EN AW 5052-Al Mg2.5 and the stainless steel is AISI 304 [31,32].

### 3.2. Intrinsic Transfer Matrix Method (ITMM)

Figure 7 shows the frequency spectrum for the standard bar. Once the natural frequency of the system was determined experimentally, the Mathcad program was used to obtain the real roots of Equation (4), indicated above. Then, the values of the sound propagation speeds for each analyzed sample were determined. Thus, the results of the acoustic and elastic properties of the two studied materials are shown in Table 3.

By comparing the results obtained from the two methods used for assessing the mechanical properties of the aluminum and stainless steel samples, it was found that there was an error of 4.87% for the longitudinal modulus of elasticity of the aluminum samples and 2.09% in case of the stainless steel samples. Comparing the results obtained experimentally with those from the references, it was found that the deviations are below 5%, which confirms that both methods presented in this research are valid. The speed of sound propagation in the two materials, determined by the intrinsic transfer matrix method, has deviations of 1.74–2.27% from the data indicated in [29,30,31,32].

### 3.3. Scanning Electron Microscopy (SEM) and Energy-Dispersive X-ray Spectroscopy (EDX) Analysis

Surface morphology of the samples and their surface chemical elemental composition (%) were determined with the aid of SEM equipped with EDS. Depending on the applications in which the aluminum alloy is used, the main alloying elements are copper, magnesium, zinc, silicon, manganese and lithium. For each material, several measurements were made on the basis of which the average value of the chemical composition was obtained (Figure 8). The EDX images of the samples are presented in Figure 8a that show the distributions of the main chemical elements in the aluminum sample surface. The percentage values of the elemental chemical composition are summarized in Table 4. Thus, it was identified that aluminum (Al) is found the proportion of 85%, carbon (C), in a proportion of 8.99%, oxygen (O), 3.97%, magnesium (Mg), 0.96% and calcium (Ca), 0.88%. The relatively large amount of oxygen is due to the oxidative process produced immediately after the breaking of the aluminum specimen. Similar data were reported by [33,34,35,36,37]. For the stainless steel sample (Figure 8b), the elemental chemical composition includes the following chemical elements: iron (Fe) in the proportion of 65.44%, carbon (C) 20.96%; oxygen (O), 10.47%; calcium (Ca), 1.84% and aluminum (Al), 1.31%, as shown in Table 4.

By analyzing the micro cross section of the specimens broken during the tensile testing, noticeable differences are observed between the breaking of the aluminum samples compared to the steel samples. It is worthy to note that the SEM images of the fractured aluminum sample, Figure 9a,b, whose structural planes are sliding over each other, thus confirm the tensile test result that the sample fracture occurred through the necking mechanism. This fracture is specific to the ductile fracture type and shows that the aluminum sample is able to sustain significant deformation or plastic strain. The SEM images obtained from the fracture zone of the stainless steel samples showed a predominantly brittle fracture, as can be seen in Figure 9c,d, with more cracking zones and pores than the aluminum sample. Similar data were reported in [38,39,40,41,42,43]. In the case of the stainless steel samples, it was observed that the fracture angle between the applied traction load axis and the fracture surface was almost perpendicular and showed a relatively flat fracture surface, as seen in Figure 9b.

### 3.4. Modal Analysis of Triangles

In this part, the values of physical and elastic properties for aluminum and stainless steel were applied in the simulation of the dynamic behavior of triangles with respect to the real sizes. By analyzing the natural frequencies, it can be seen that the first mode of vibration occurs in the plane of the triangle (transverse mode), at a frequency of 156 Hz. The second mode, at 162 Hz, is a vibration mode that occurs through the vibrations of the free sides of the triangle outside the plane of the triangle, also called the torsion mode. The third mode of vibration, produced at a frequency of 269 Hz, is manifested by longitudinal vibrations of the triangle. These are the main vibration modes of the triangle, the following similar modes, occurring at higher frequencies. Regarding the dominant frequency (Figure 10), it is found that the aluminum triangle has the highest dominant frequency (1857 Hz—Figure 10a), which is in good agreement with the sound velocity determined above (5089 m/s), compared to the stainless steel triangle (1645 Hz—Figure 10b), whose sound velocity is 4889 m/s, Table 3. Increasing the sides of the aluminum triangle reduces the dominant frequency by about 43% compared to the triangle with smaller sides. Figure 10 shows the vibration modes of the dominant frequencies of the stainless steel and aluminum triangles obtained from the numerical analysis. It can be noticed that the nodes and antinodes of the modal shapes of those triangles are similar.

## 4. Conclusions

The paper investigated the mechanical and acoustic properties of two types of metal-based materials used in the construction of percussion musical instruments, one based on aluminum and the other on stainless steel. In this regard, two experimental methods were applied, both leading to the same results, which confirms that the characterization of the materials was correct.

The main findings of this study are:Identification of the elastic, acoustic and morphological properties of the materials used in the construction of musical triangles;Convergence of the experimental results obtained by the two methods, with an error of less than 5%;Chemical and morphological analysis confirmed the ductile and brittle fracture pattern of the aluminum and stainless steel materials, respectively;According to the mechanical and acoustic test results, the aluminum-based musical triangle could be used by players for high crystalline sound, while the stainless steel musical triangle could be used when lower frequencies are required.

In future studies, other types of materials that can be used in the construction of percussion instruments will be analyzed and some solutions to improve their acoustic properties.

## Figures and Tables

**Figure 1 materials-15-05192-f001:**
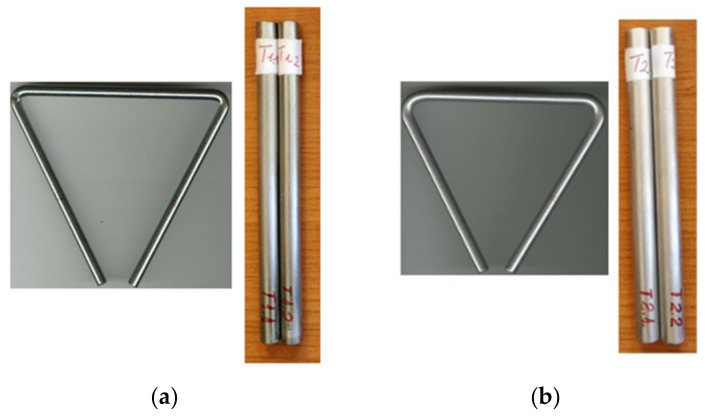
Types of samples for the tensile test: (**a**) stainless steel samples coded T1 (respectively T1.1 and T1.2); (**b**) aluminum sample coded T2 (respectively T2.1 and T2.2).

**Figure 2 materials-15-05192-f002:**
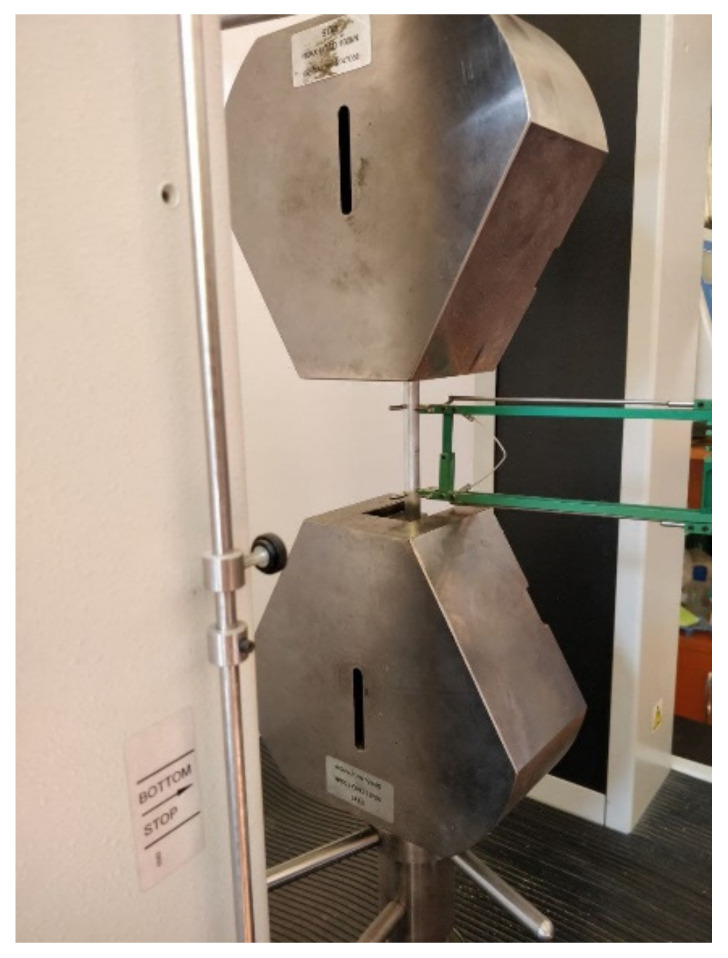
The tensile test.

**Figure 3 materials-15-05192-f003:**
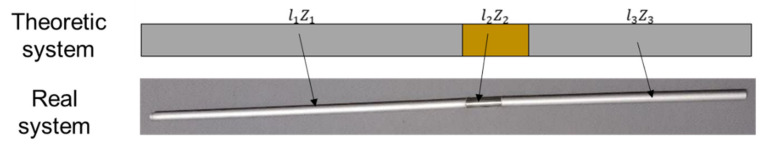
The ternary system. Legend: $l_{1}$,$\;l_{2},\;\;l_{3}$ represent the length of each rod; $Z_{1}$, $Z_{2},$ $Z_{3}$ is the acoustic impedance of rods.

**Figure 4 materials-15-05192-f004:**
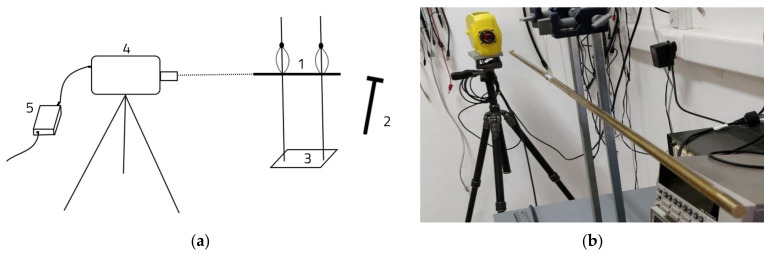
The experimental set-up: (**a**) the principle of experimental determination of the natural frequency of the standard bar used for the intrinsic transfer matrix. Legend: 1—sample; 2—impact hammer; 3—holder; 4—vibrometer; 5—data acquisition device; (**b**) the experimental stand.

**Figure 5 materials-15-05192-f005:**
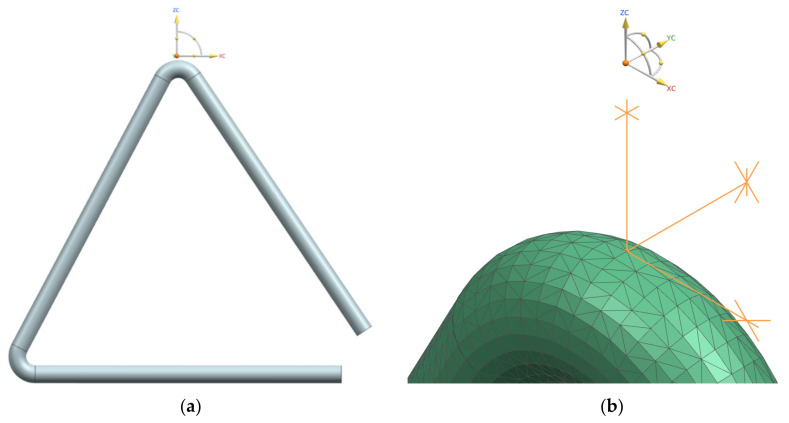
The modeling of the triangle using finite element analysis: (**a**) geometry of triangle; (**b**) meshing of the geometric model and blocked node.

**Figure 6 materials-15-05192-f006:**
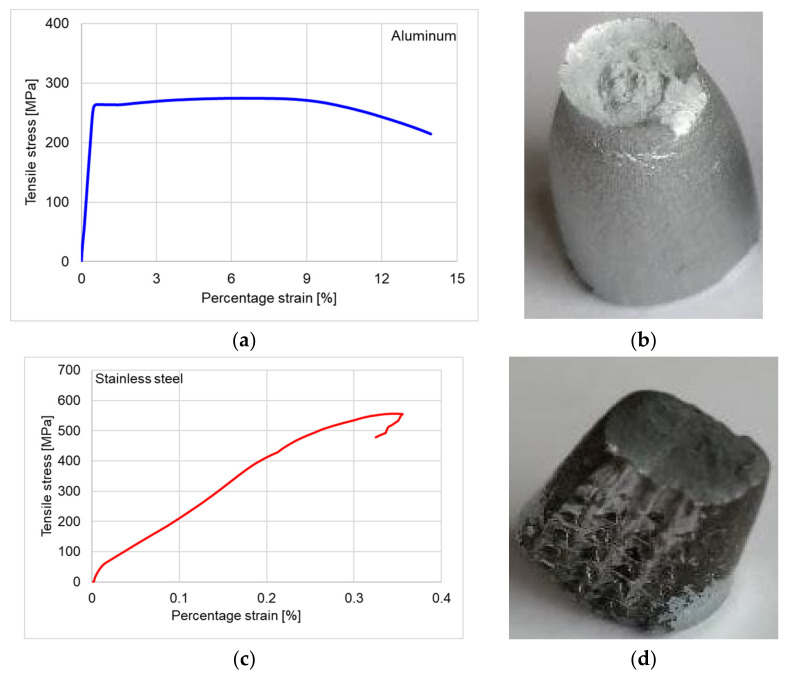
The tensile stress–strain curves: (**a**) aluminum curve; (**b**) necking behavior of aluminum sample; (**c**) stainless steel curve; (**d**) brittle fracture of stainless steel sample.

**Figure 7 materials-15-05192-f007:**
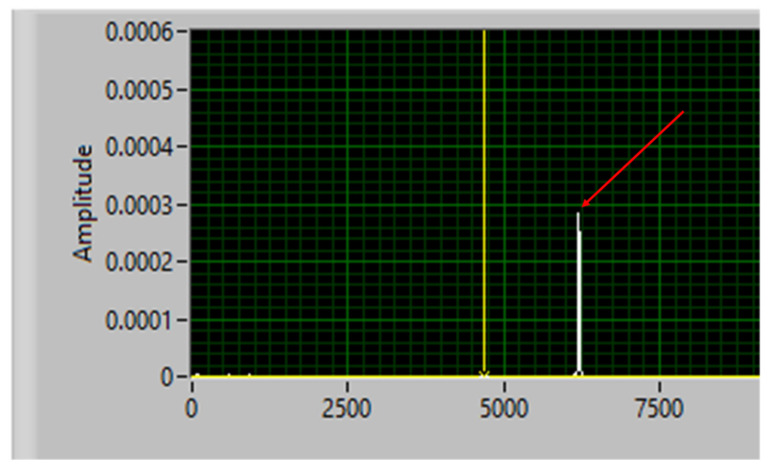
The natural frequency of the standard bar in the ternary system. Legend: the red arrowhead indicates the natural frequency $f_{n}$ of 6112.42 Hz.

**Figure 8 materials-15-05192-f008:**
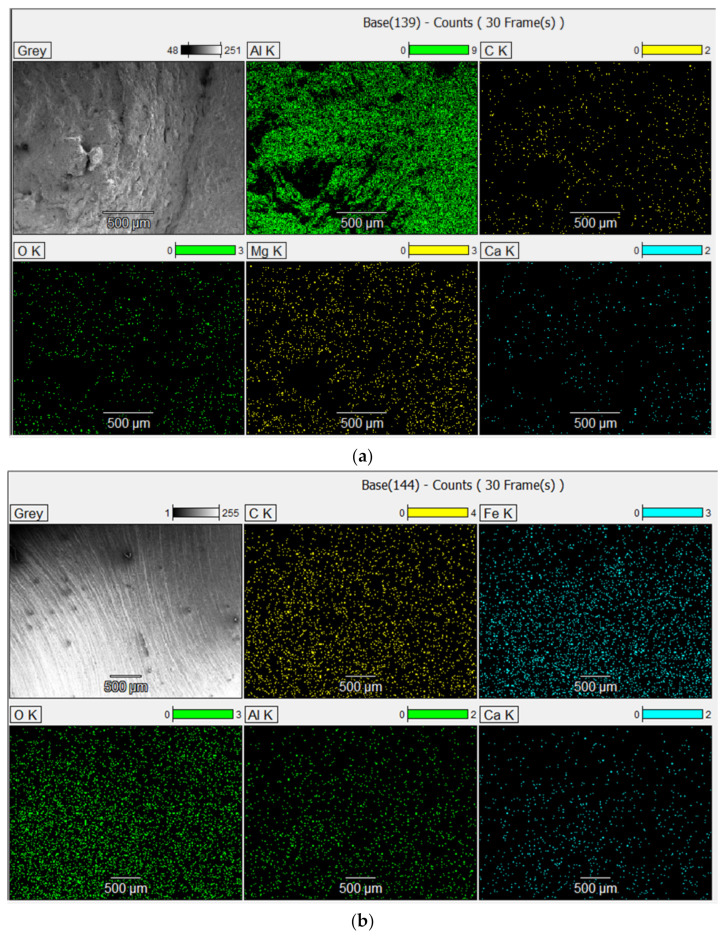
EDX mapping fracture image of: (**a**) the aluminum sample; (**b**) stainless steel sample.

**Figure 9 materials-15-05192-f009:**
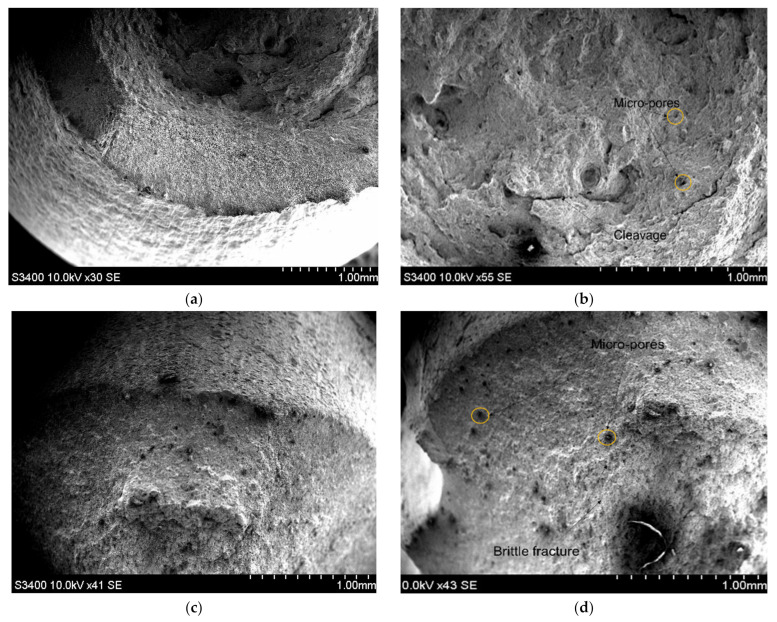
SEM images obtained after the tensile test in the fractured zone of: (**a**) aluminum sample (**b**) aluminum sample; (**c**) stainless steel sample; (**d**) stainless steel sample.

**Figure 10 materials-15-05192-f010:**
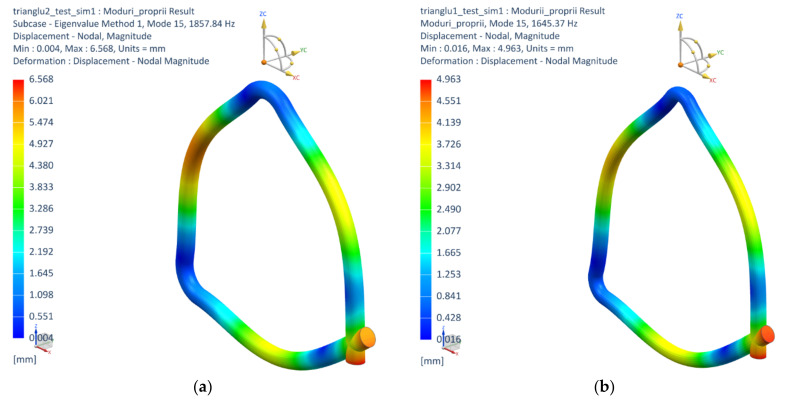
The vibration modes corresponding to dominant frequencies of those triangles: (**a**) stainless steel triangle; (**b**) aluminum triangle.

**Table 1 materials-15-05192-t001:** The physical features of the samples.

Type of Material	Length (mm)	Diameter (mm)	Density (kg/m^3^)	The Experimental Method
Aluminum	182.21	9.593	2708	Tensile test
Stainless steel	118.50	7.860	7818	Tensile test
Aluminum	40.00	9.593	2708	Intrinsic transfer matrix
Stainless steel	40.00	7.860	7818	Intrinsic transfer matrix

**Table 2 materials-15-05192-t002:** The mechanical properties of aluminum and stainless steel obtained by tensile test.

Type of Material	Modulus of Elasticity (GPa)	Tensile Strength (MPa)
Aluminum	66	273
Stainless steel	182	555

**Table 3 materials-15-05192-t003:** Comparison between the mechanical properties of aluminum and stainless steel obtained by experimental methods (Legend: ITMM—intrinsic transfer matrix method; TT—tensile test; Ref—reference).

Type of Material	Sound Propagation Velocity (m/s)		Difference %	Modulus of Elasticity (GPa)			Difference %		
	ITMM	Ref. [27]		ITMM	TT	Ref. [28]	ITMM-TT	ITMM-Ref. [28]	TT-Ref. [28]
Aluminum	5089	5000	1.74	70	66.59	69	4.87	1.45	3.49
Stainless steel	4889	5000	2.27	186	182.12	180	2.09	3.33	1.18

**Table 4 materials-15-05192-t004:** The chemical composition of the samples alloys.

Type of Material	Elemental Chemical Composition				
Aluminum	0.02% C	1.57% O	0.96% Mg	85.44% Al	0.80% Ca
Stainless steel	0.08% C	0.47% O	1.31% Al	65.66% Fe	1.84% Ca

## Data Availability

Not applicable.

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
