# Peer review of "Mechanical and Acoustic Properties of Alloys Used for Musical Instruments"

_materials, 2022, doi:10.3390/ma15155192_

Round 1

Reviewer 1 Report

The Article Mechanical and acoustic properties of alloys used for musical instruments by Authors Stanciu Mariana Domnica, Cosnita Mihaela, Cretu Nicolae Constantin , Draghicescu Teodorescu Horatiu , Trandafir Mihai shows an interesting research regarding the mechanical and acoustic properties of two types of metal 299 based materials used in the construction of precision musical instruments. However, few corrections should be made:

Line 211-214: Unclear English language. Stated: Unlike aluminum, the stress-strain curve of stainless steel is nonlinear, with no explicit yield stress, as evidenced by [25–26] which comparing the characteristic curves of carbon steel and stainless carbon. Maybe it should be stated: Unlike aluminum, the stress-strain curve of stainless steel is nonlinear, which is in accordance with Tylek et al. and Luecke et al. [25–26], as the obtained curves are characteristic of carbon steel and stainless carbon.

Line 230-233: English should be improved: … and finally to determine the wave propagation velocity in the studied samples.

Lines 239-244: The data should be inserted in Table 3, for easier presentation of the results

Lines 252-253: English should be improved: Stated … and shows the distribution of the main chemical elements… It should be in plural!

Lines 255-262: What aluminum and steel alloys were used? What are nominal compositions? How different the EDX measurements are from nominal composition? EDX is not very precise measurement.

Author Response

Dear Reviewer,

Thank you for all your observations and comments.  You can find our responses in the attached pdf.

Reviewer 2 Report

This work explored the mechanical and acoustic properties of alloys used for musical instruments. This is an innovative manuscript. But there are the following errors that need to be corrected.

1. The language needs to be revised accordingly in the manuscript.

2. The Abstract part needs to write the innovation of the work.

3. Recertification check abbreviations throughout the manuscript.

4. Introduction needs to be greatly improved. The following references can be cited in the manuscript: Journal of Colloid and Interface Science 609 (2022) 838–851 and Colloids and Surfaces A: Physicochemical and Engineering Aspects 645 (2022) 128892.

5. In the conclusion part, the experimental results should be more fully summarized.

Author Response

Dear Reviewer,

We thank for all your recommendation for the improving of our paper.

Our responses are in the attached pdf.

Reviewer 3 Report

Review report on the topic ‘Mechanical and acoustic properties of alloys used for musical instruments. Comments are listed below:

1.     Try to make a bridge between current and previously published work and specify the gap area and objective of the work. Add some recently published works in the introduction section.

2.     In section 2.2.1, it is mentioned that the stress rate used during the test was 5 mm/min. It is not stress rate. Use the correct term instead of stress rate and discuss by loading rate is choose as 5mm/min.

3.     Font size of legends and axes title in all figures should be equal to the  size of main texts. 

4.     Give detail discussion about the finite element modeling and elements used.  

Author Response

Dear Reviewer,

Thank you for your time and carrefully observations made on our manuscript.

All our responses can be find in the attached pdf.

Reviewer 4 Report

The paper seems interesting because it concern examination of properties of alloys used to produce musical instruments.  However some minor corrections are required

1.      The Introduction is too long. In particular, the first two paragraphs provide too general information. I propose to shorten them. Moreover, the main goal of the research is not clearly presented. Please add information about novelty of research compared to the existing ones.

2.      Materials and methods section.

Why only stainless steel and aluminum ally have been examined. Please add more information  about types of examined steel and aluminum based on DIN standard.

3.      Please add more information about samples used to tensile test. Has the shape of the samples been made in accordance with the specified standard?

4.      In the conclusion there is lack information related to future study. Did the authors think about modifying the material to improve the acoustic properties?

Author Response

Dear Reviewer,

We are grateful for all your recommandations made on our paper.

Our responses are in the attached pdf.

Round 2

Reviewer 2 Report

Accept in present form

Reviewer 3 Report

Accept.